# Novel Quantitative PCR for *Rhodococcus equi* and Macrolide Resistance Detection in Equine Respiratory Samples

**DOI:** 10.3390/ani12091172

**Published:** 2022-05-03

**Authors:** Sonsiray Álvarez Narváez, Ingrid Fernández, Nikita V. Patel, Susan Sánchez

**Affiliations:** 1Poultry Diagnostic and Research Center, Department of Population Health, College of Veterinary Medicine, University of Georgia, Athens, GA 30602, USA; 2Athens Veterinary Diagnostic Laboratory, Department of Infectious Diseases, College of Veterinary Medicine, University of Georgia, Athens, GA 30602, USA; ingridf@uga.edu (I.F.); nikita.patel2@uga.edu (N.V.P.); ssanchez@uga.edu (S.S.)

**Keywords:** *Rhodococcus equi*, qPCR, diagnosis, macrolide resistance, *erm*(46), *erm*(51)

## Abstract

**Simple Summary:**

Multidrug-resistant (MDR) *Rhodococcus equi* (*R. equi*) is rapidly spreading across the US in equine breeding farms, causing devastating untreatable disease in foals. There is a critical need for new diagnostic tools that can identify *R. equi* and its antibiotic resistance profile accurately and fast. This project aimed to develop and standardize a novel multiplex quantitative real-time PCR (qPCR) to detect *R. equi* and its most clinically relevant antimicrobial resistance genes directly from equine respiratory samples. We have designed three pairs of oligos (a.k.a primers or oligonucleotides) to identify *R. equi* and MLS_B_ (macrolide, lincosamide, streptogramin B) resistance genes *erm*(46) and *erm*(51) that can be used in single-plex and multiplex qPCR assays. Furthermore, our new qPCR shows high sensitivity and specificity in in-silico analysis and when tested in mock equine respiratory samples. Therefore, we believe it can be used for a fast-preliminary diagnosis of *R. equi* and the simultaneous prediction of its most critical resistant profile. The new molecular diagnostic tool presented here will shorten the waiting time from the moment the practitioner sees the equid patient until it is diagnosed and appropriately treated.

**Abstract:**

*R. equi* is an important veterinary pathogen that takes the lives of many foals every year. With the emergence and spread of MDR *R. equi* to current antimicrobial treatment, new tools that can provide a fast and accurate diagnosis of the disease and antimicrobial resistance profile are needed. Here, we have developed and analytically validated a multiplex qPCR for the simultaneous detection of *R. equi* and related macrolide resistance genes in equine respiratory samples. The three sets of oligos designed in this study to identify *R. equi* housekeeping gene *choE* and macrolide resistance genes *erm*(46) and *erm*(51) showed high analytic sensitivity with a limit of detection (LOD) individually and in combination below 12 complete genome copies per PCR reaction, and an amplification efficiency between 90% and 147%. Additionally, our multiplex qPCR shows high specificity in in-silico analysis. Furthermore, it did not present any cross-reaction with normal flora from the equine respiratory tract, nor commonly encountered respiratory pathogens in horses or other genetically close organisms. Our new quantitative PCR is a trustable tool that will improve the speed of *R. equi* infection diagnosis, as well as helping in treatment selection.

## 1. Introduction

*R. equi* is an animal and human pathogen, well known in veterinary medicine as the leading cause of severe bronchopneumonia in foals [1]. The emergence and spread of MDR *R. equi* represent a significant threat to the equine industry due to the economic impact associated with this disease, related to animal long-term treatment costs, veterinary and nursing care costs, and lost income from animal mortality. MDR *R. equi* is also a concern to public health as the antibiotic therapy of choice to treat *R. equi* infections in horses (a macrolide in combination with rifampin) is also routinely used to treat bacterial infections in humans [2,3].

Our previous investigation determined that in the US, *R. equi* resistant to macrolides and rifampin mainly cluster in three clonal populations: clone 2287, clone G2016, and clone G2017 [4,5,6]. Clones 2287 and G2016 carry the MLS_B_ resistance gene *erm*(46) as a part of transposon tn*RErm46,* and a different rifampin resistance based on point mutations in the *rpoB* gene at position 531 [4,7]. *R. equi* clone G2017 carries a different MLS_B_ resistance gene, *erm*(51) in transposon tn*RErm51* [6], and a *rpoB* mutation in position 531, which confers rifampin resistance [6].

Due to the insidious progression of infection to severe clinical signs, early and accurate diagnosis of foals with *R. equi* pneumonia is essential. We present a new qPCR approach to quickly and efficiently identify macrolide-resistant *R. equi* directly from equine respiratory samples.

## 2. Materials and Methods

### 2.1. Bacterial Genomes

The 202 *R. equi* genomes included in our analysis were randomly selected Illumina whole-genome assemblies from isolates characterized in previous studies (see GeneBank accession numbers in Appendix A): 105 environmental genomes (*n* = 35 macrolide susceptible and *n* = 70 macrolide resistant) from Huber et al., 2020 [6]; 62 clinical isolates (*n* = 19 macrolide susceptible and *n* = 43 macrolide resistant) from Álvarez-Narváez et al., 2019 [7]; and 2021 [5]; and 35 macrolide susceptible strains representative of the global genomic diversity of *R. equi* [8]. The 25 non-*R. equi* reference genomes were selected based on genetic similarity with *R. equi* and previous literature [9,10].

### 2.2. Oligo and Probe Design and In-Silico Validation

Genes *choE, erm*(46), and *erm*(51) were extracted from the genomes of *R. equi* 103S (NCBI accession number NC_014659.1), *R. equi* PAM2287 (NCBI accession number ASM209440v1), and *R. equi* lh_50 (NCBI accession number ASM1687895v1) using Artemis software (v18.1.0) [11]. Oligos and probes were designed using the function “find primers” from ApE-A plasmid Editor (v2.0.61, [12]). Geneious mapper (Version 6.0.3, [13]), set up with a high sensitivity method and fine-tuned with up to five iterations, was used to map 202 *R. equi* genomes (Appendix A) and 25 non-*R. equi* reference genomes (plasmids and chromosomes) to the three sets of oligos and probes designed in this study. Single nucleotide polymorphisms (SNPs) between the oligo sets and the tested sequences were quantified by hand from the sequence alignment. The expected amplification detection was calculated based on QuantiNova DNA polymerase processivity (Taq DNA Polymerase 2–4 kb/min) and PCR elongation time (30 s). This way, amplification would only be detected if the forward and reverse primers are in <2 kb proximity and the corresponding probe is inside that <2 kb DNA fragment.

### 2.3. Bacterial Strains and Culture Conditions

Three *R. equi* strains were used in this study: *R. equi* PAM2287 (NCBI accession number ASM209440v1), a macrolide and rifampin-resistant clinical isolate that carries the virulence plasmid pVAPA, the macrolide resistance gene *erm*(46), and the rifampin mutation *rpoB^S53F^* [7]; *R. equi* 103-Apra^R^ is a plasmidless derivative strain of *R. equi* 103 containing the *aac(3)IV* apramycin resistance cassette integrated into the chromosome [14]; and R. equi lh_50 (NCBI accession number ASM1687895v1) is a macrolide and rifampin-resistant environmental isolate carrying the macrolide resistance gene erm(51), and the rifampin mutation rpoBS531Y [6]. All R. equi strains were routinely cultured in a brain–heart infusion medium (BHI; Difco Laboratories-BD) at 37 °C and 200 RPMs unless otherwise stated. Agar media were prepared by adding 1.6% of bacteriological agar (Oxoid). When required, media was supplemented with antibiotics (erythromycin, 8 µg/mL; and rifampin, 25 µg/mL; Sigma- Aldrich, St. Louis, MO, US). Additionally, qPCR specificity was tested on 12 wildtype *R. equi* (Appendix A) and five clinical isolates frequently isolated from the respiratory tract of healthy and sick horses (*Corynebacterium pseudotuberculosis*, *Streptococcus equi* subsp. *equi*, *Streptococcus equi* subsp. *zooepidemicus*, *Nocardia asteroides*, and *Mycobacterium avium* subsp. *paratuberculosis*) from the Athens Veterinary Diagnostic Laboratory (Athens, GA, USA) collection, extracting DNA directly from frozen stocks.

### 2.4. DNA Extraction and Real-Time qPCR Assay

DNA was extracted from frozen stocks, isolated colonies, and equine nasal swabs using IndiSpin^®^ Pathogen Kit (Indical Bioscience, Orlando, FL, US) following manufacturer’s instructions. qPCRs were carried out using the QuantiNova Pathogen +IC Kit (Qiagen, Germantown, MD, US) in a 7500 Real-Time PCR System thermocycler (Applied Biosystems, Bedford, MA, US). For single qPCRs, a master mix volume of 20 μL per reaction was used mixing 5 μL of DNA template, 5 μL of Quantinova Mix, 0.1 μL of Quantinova Rox, 2 μL of each forward and reverse 10 μM primers (Table 1), 1 μL of corresponding 6 μM probe (Table 1), and 4.90 μL of molecular grade water (Fisher). For multiplex qPCR, a master mix volume of 20 μL per reaction was used mixing 5 μL of DNA template, 5 μL of Quantinova Mix, 0.1 μL of Quantinova Rox, 1 μL of each forward and reverse 10 μM primer combination (*n* = 6 primers) (Table 1), and 1 μL of each of the three 6 μM probes. PCR reactions were performed in triplicate. The thermocycler conditions used for both singleplex and multiplex were 5 min at 95 °C of initial denaturation, and 40 cycles of amplification with 5 s at 95 °C of denaturation, followed by 30 s of oligonucleotide hybridization at 60 °C, and 30 s of elongation at 68 °C. CT (cycle threshold) values were estimated automatically by QuantStudio^TM^ Design and Analysis Software (v 1.5.1, Applied Biosystems, Bedford, MA, US). CT cut off for all our qPCRs was set at 40 cycles, and CT values below 35 were call positive; CT values between 35 and 40 were called suspect; and Ct values after 40 were considered negative. Cut offs were estimated based on limits of detection (LOD) results. DNA from *R. equi* 2287, *R. equi* 103-ApraR, and *R. equi* lh_50 was used as a positive control for the Rhodo_Dlab set of oligos. *R. equi* 2287 DNA was used as a positive control for the Erm46_Dlab set and *R. equi* lh_50 was used as a positive control for the Erm51_Dlab set. The genomes of the three strains have been fully sequenced and are publicly available in NCBI. Water (instead of bacterial DNA) was used as a universal negative control in all assays, and DNA extracted from nasal swabs of healthy horses was additionally used as an extra negative control in the multiplex qPCR in mocking samples.

### 2.5. Standard Curve Construction

*R. equi* DNA concentration was measured with a Qubit Fluorometer (ThermoFisher, Waltham, MA, US) and Qubit dsDNA BR Assay Kits (ThermoFisher, Waltham, MA, US) and adjusted to 10–15 ng/μL. Ten-fold serial dilutions were made with *R. equi* extracted DNA in molecular grade water (Fisher Scientific, Pittsburgh, PA, US). The qPCR assay stated above was used to determine the CT values for each dilution. A standard regression curve was constructed with QuantStudio^TM^ Design and Analysis Software (v 1.5.1, Applied Biosystems, Bedford, MA, US) using linear regression analysis of the log_10_ sample quantity and the corresponding CT values. Slope and regression equations were calculated for all primer sets individually and in combination. The percentage of efficiency of the qPCR reaction was calculated using the following formula: Efficiency% = (−1 + 10^(−1/slope)^) × 100.

### 2.6. Limit of Detection (LOD) Calculation

LOD is presented as the minimum number of target copies in a sample that can be measured accurately [15]. The LOD was calculated with The Rhode Island Genomics and Sequencing Center online calculator (https://cels.uri.edu/gsc/cndna.html, accessed on 22 April 2022, Andrew Staroscik, Kingston, RI, US). This calculator requires the user to input the minimum amount of template detected (MAT) in ng and the length of the template (LT) in bp. With this information the number of copies is calculated with the following formula: number of copies = (MAT_ng_ × 6.022 × 10^23^ molecules/mole)/(LT_bp_ × 1 × 10^9^ ng/g × 650 g/mole of bp). MAT was calculated with the following formula: MAT = DNA concentration of template (ng/µL) × last fold dilution detected × vol. template. *R. equi* genome size (~5.2 Mbp) was used as LT in all LOD calculations.

## 3. Results

### 3.1. Oligo Set Design and In-silico Validation

We designed for this study a total of three oligo pairs and their corresponding probes (Table 1) targeting *R. equi* housekeeping gene *choE* (Rhodo_Dlab set)*,* and macrolide resistance genes *erm*(46) (Erm46_Dlab set) and *erm*(51) (Erm51_Dlab set). We performed a preliminary in-silico validation of the oligos and probes to test their specificity following the inclusivity and exclusivity criteria [16]. We checked in-silico inclusivity by mapping the three primer sets and probes to the whole genome sequences of 202 *R. equi* (Appendix A) isolated from different animal species and presenting multiple susceptibility profiles (Table 2). The Rhodo_Dlab set mapped with the chromosomes of all the 202 *R. equi* tested at the targeted place. However, three *R. equi* genomes showed >1 SNP difference with the Rhodo_Dlab set, and to be conservative, we predicted PCR products in 199 out of 202 reactions (98.5%) (Table 2). Erm46_Dlab and Erm51_Dlab sets had perfect matches (zero SNPs) with the genomes of all *erm*(46)-positive (*n* = 85) and *erm*(51)-positive (*n* = 29) *R. equi*, respectively, and PCR products were predicted for all reactions (Table 2). Although we observed that the three oligo sets had matches with non-targeted *R. equi* sequences, amplification was never expected as the combination of the two oligos and corresponding probe was never found in enough proximity to generate a PCR product (Table 2). Additionally, all non-target matches were full of SNPs (data not shown) and that would most probably hamper the alignment between the oligos/probe and the *R. equi* genome.

We tested the in-silico exclusivity by mapping the three primer sets and probes to the whole genome sequences of 25 non-*R. equi* reference genomes (plasmids and chromosomes) of bacteria species in close genetic proximity with *R. equi* and other common respiratory pathogens. Similar to the observations made on the non-targeted *R. equi* sequences, occasionally, one of the oligos or the probe would align with non-*R. equi* genomes. (Appendix A). Still, detection of amplification was never expected as none of the two primers and corresponding probes align in the same genome simultaneously. (Table 2 and Appendix A).

### 3.2. Testing Oligos and Probes in Singleplex and Multiplex qPCR Assays

We first explored the optimal hybridization temperature for each pair of oligos in a conventional singleplex PCR with a temperature gradient between 55 °C and 70 °C (Figure 1). We observed that the three oligo pairs produced the desired band size (~200 bp) regardless of the hybridization temperature used. Still, their performance was better between 60–70 °C. Most of the qPCR diagnostic tests run in our laboratory use 60 °C as a hybridization temperature. Hence, we decided to use 60 °C as a hybridization temperature in this assay for convenience.

Then we tested the qPCR efficiency and analytic sensitivity (expressed as the limit of detection [LOD]) for each oligo set individually. Macrolide resistant *R. equi* 2287 was used as a template to characterize the Rhodo_Dlab set and the Erm46_Dlab set, while *R. equi* lh_50 was used with the Erm51_Dlab group. PCR efficiencies ranged between 104 and 121%, with coefficients of determination (R^2^) above 0.99 (Table 3. The qPCR detected *R. equi* (gene *choE*) even when just as little as 6 × 10^−^^5^ ng of *R. equi* DNA was present in the sample (Figure 2 and Appendix A), and the LOD for the Rhodo_Dlab set was estimated to be 10.7 complete genome copies per PCR reaction (Table 3). Similarly, the oligo sets designed to identify macrolide resistance genes *erm*(46) and *erm*(51) detected the corresponding targets, even when a small load of *R. equi* DNA was present in the sample (~6 × 10^−^^6^ ng, Figure 2 and Appendix A). The LOD for the Erm46_Dlab and Erm51_Dlab oligo sets was estimated to be 1.18 and 1.07 complete genome copies per PCR reaction, respectively (Table 3).

Finally, we tested the performance of each oligo pair when working together in a multiplex qPCR assay. This time, we decided to use a pool of the three *R. equi* strains mentioned above (ratio 1:1:1) as a DNA template. We observed that the qPCR efficiency and R^2^ value decreased for all primers sets (Table 3). No changes in efficiency and sensitivity were observed for the Rhodo_Dlab pair (Table 3). Similarly, a 10-fold decrease in LOD was observed for Erm46_Dlab and Erm51_Dlab oligo sets, and no changes were observed for the Rhodo_Dlab set.

### 3.3. Testing the Analytic Sensitivity and Specificity in Mocking Equine Respiratory Samples

We extracted and pooled the microbial DNA content of 13 nasopharyngeal swabs from 13 healthy horses (one swab per animal). We used that as representative DNA of the normal equine respiratory flora. We used this DNA extract to dilute *R. equi* DNA (pool of the three *R. equi* isolates mentioned above) in 10-fold serial dilutions to have a “mock” respiratory sample containing decreasing amounts of *R. equi* DNA, and we recalculated the multiplex qPCR efficiency and analytic sensitivity (LOD) (Table 3). We observed that no CT values were recorded for the respiratory microbial DNA when *R. equi* DNA was not present (Figure 3), demonstrating that our multiplex qPCR is not reactive to normal bacterial communities present in the respiratory tract of horses. Although we found that the LOD slightly decreased in this assay compared to the multiplex qPCR performed on clean *R. equi* DNA, the amplification efficiency did not significantly change (Table 3).

Additionally, we determined the analytic specificity of the multiplex qPCR by testing its inclusivity, or the ability to detect a wide range of *R. equi* isolates of the ADVL collection (*n* = 12, Appendix A), and its exclusivity, or lack of interference from three genetically similar organisms *C. pseudotuberculosis, N. asteroids*, and *M. avium* subsp. *paratuberculosis,* and two common equine respiratory pathogens, *S. equi* subsp. *equi, S. equi* subsp. *zooepidermicus* (Figure 3). The multiplex qPCR detected all the *R. equi* isolates (Appendix A). At the same time, no signal (CT values) was observed for the non-*R. equi* species for any oligo pairs (Figure 3), proving that our strategy is specific for *R. equi* and not reactive to non-target DNA. All wildtype clinically obtained *R. equi* tested in this assay were susceptible to macrolides. No signal was detected for the Erm46_Dlab and Erm51_Dlab oligo sets (Appendix A), further demonstrating specificity.

## 4. Discussion

The current diagnosis for *R. equi* pneumonia in foals generally involves a cytology report describing the presence of pleomorphic rods in respiratory fluids, a preliminary qPCR detecting *R. equi* directly from the respiratory sample, and subsequent bacteria isolation and antimicrobial susceptibility profiling [17]. Cytology and PCR results are usually available to the clinician in the first 12 h. However, *R. equi* isolation and corresponding antimicrobial susceptibility testing can take up to 72 h. Due to the insidious nature of this disease [1], fast action is crucial, and clinicians generally start treatment with a macrolide (clarithromycin) and rifampin before the susceptibility profiles are available. This strategy has been effective so far as resistance to macrolides and rifampin was below 5% of all *R. equi* cases recorded in some studies [5,18]. However, over the last 15 years, the number of resistant isolates has significantly increased, to the point that resistant *R. equi* to all macrolides and rifampin is being cultured from up to 40% of foals at a farm in Kentucky [19]. The increasing prevalence of MDR *R. equi* in the US has been recently evidenced by epidemiological studies such as Dr. Huber’s et al. in Kentucky [18,20], and requires the development of new diagnostic tools like the qPCR described here, that account for bacterial detection and clinically relevant resistance identification directly from respiratory samples, fast and accurately.

The dual antimicrobial therapy to treat *R. equi* infections in horses mentioned above has been used for over 30 years [18,21,22,23]. In the US, resistant *R. equi* are mainly grouped in three clonal populations with entirely different genetic backgrounds, although they share similar antimicrobial resistance mechanisms [4,5,6]. All resistant clones present a rifampin resistance point mutation at position 531 of the *rpoB* gene, and they all carry an MLS_B_ resistance *erm* gene inserted in a transposon [4,5,6]. So far, two different *erm* genes have been identified in *R. equi*, *erm*(46) primarily found in clinical isolates (clones 2287 and G2016) [4,24], and *erm*(51) mainly associated with environmental samples [6]. Although *erm*(46) and *erm*(51) appear to be part of mobilizable elements that could mobilize into other bacteria through horizontal gene transfer [6,7,25], these genes have only been observed in the genomes of *R. equi*. Additionally, genetic analysis showed that *erm*(46) and *erm*(51) share low similarity with each other (~50% at nucleotide level [6]) and with other known *erm* genes (<68% nucleotide sequence identity to *erm*(38), *erm*(39) and *erm*(40) found in *Mycobacterium* spp.) [6]. Our multiplex qPCR targets these two *erm* genes and not the rifampin resistance as punctual mutations are challenging to identify by qPCR, and as epidemiology evidence shows, if an isolate is identified as *R. equi* and carries an *erm* gene, it most probably also contains a rifampin resistance mutation [4,5,6].

Like previous approaches, our qPCR strategy targets the *R. equi* housekeeping gene *choE* to identify *R. equi* [10,26,27,28]. Still, we are using a new oligo combination that allows the assay to be run in multiplex with two more sets of primers to detect *erm*(46) and *erm*(51) genes. We decided not to target the virulence gene *vapA* as some did previously [10,26,29,30,31] as *vapA* is part of a plasmid, and there is a slight chance that this virulence gene is not present in equine *R. equi* isolates [28,32]. We first performed an in-silico validation of our assay to have preliminary data supporting the specificity of our primers and probes. In-silico validations of molecular-based detection methods are gaining popularity [33,34,35]. However, to our knowledge, this is the first time this approach has been used to develop a qPCR to detect *R. equi.* The Rhodo_Dlab set showed high in-silico specificity as it mapped with all the chromosomes of the 202 *R. equi* tested and did not show significant matches with any of the non-*R. equi* genomes (Appendix A).

Still, a small proportion of *R. equi* genomes (22/202) showed SNPs with the Rhodo_Dlab oligo set: 19 genomes presented only one SNP difference with the Rhodo_Dlab set and three *R. equi* genomes showed >1 SNP (Table 2). SNPs in primer and probe binding sites can destabilize oligonucleotide binding and reduce target specificity [36]. Yet, PCR performance studies showed that a single SNP does not prevent amplification [37]. Hence, we estimated that the in-silico inclusivity of the Rhodo_Dlab set is 98.5%, and its exclusivity is 100%. Erm46_Dlab and Erm51_Dlab sets only had perfect matches (zero SNPs difference) with the genomes of macrolide-resistant *erm*(46)-positive and *erm*(51)-positive *R. equi,* respectively. They were predicted to have 100% inclusivity and exclusivity (Table 2). Overall, the in-silico validation of the three primer sets designed in this study showed that they are highly specific to *R. equi* in the case of the Rhodo_DLab set and macrolide resistance genes *erm*(46) and *erm*(51) (Erm46_Dlab and Erm51_Dlab oligo sets, respectively). This encourages us to continue testing our qPCR assay in-vitro.

The LOD and amplification efficiency of our *choE* oligo set (Rhodo_DLab) in singleplex and multiplex assays are comparable to what was previously published [26], and the three oligo sets used in this study showed LODs and amplification efficiencies above what considered an optimal performance. Additionally, we proved the high analytic specificity of this new *R. equi* multiplex qPCR by testing its inclusivity and exclusivity [16]. In total, 100% of the *R. equi* strains tested were identified, while none of the non-target species gave a signal in the multiplex. Furthermore, our multiplex qPCR showed not to be reactive to normal bacterial communities present in the respiratory tract of horses, further demonstrating the utility of our assay as a crucial *R. equi* diagnostic tool. Unfortunately, all the *R. equi* clinical isolates of the AVDL collection were susceptible to macrolides, so we could not test inclusivity. Future work will focus on validating this qPCR with respiratory samples from horses diagnosed with pneumonia caused by *R. equi* susceptible and resistant to macrolides and rifampin. Additionally, new resistances to macrolides and rifampin that our test does not detect may develop. Hence, although our qPCR strategy is an excellent tool to provide a fast answer to clinicians, verification based on bacterial culture and susceptibility testing for all samples is paramount, even if tested by this method.

## 5. Conclusions

This manuscript describes a new multiplex qPCR assay for the simultaneous identification of *R. equi* and its two most clinically relevant macrolide resistance genes, *erm*(46) and *erm*(51). Our qPCR approach is highly specific and sensitive, even when performed directly from equine respiratory samples.

## Figures and Tables

**Figure 1 animals-12-01172-f001:**
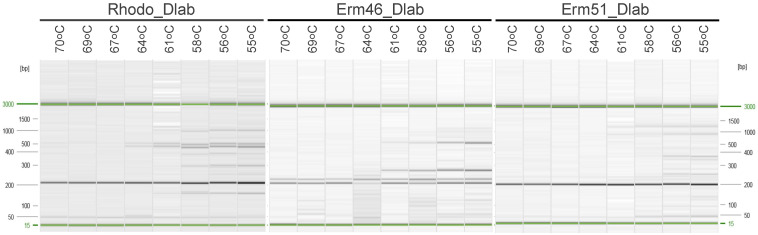
Conventional PCR with temperature gradient for oligo sets Rhodo_Dlab, Erm46_Dlab, and Erm51_Dlab. Expected band of ~200 bp size.

**Figure 2 animals-12-01172-f002:**
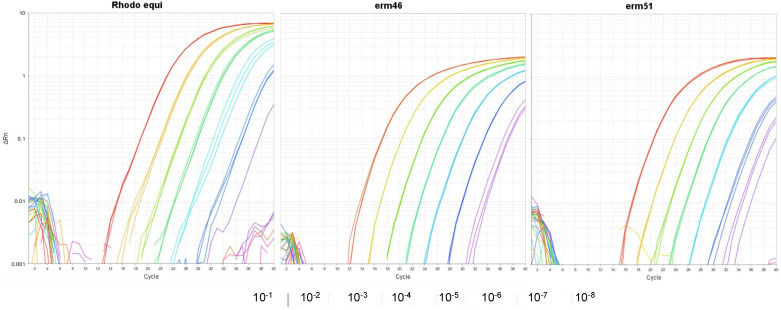
Testing the analytic sensitivity of Rhodo_Dlab, Erm46_Dlab, and Erm51_Dlab sets in singleplex. qPCR curves for the three oligo sets in singleplex assays. In the Y-exe ΔRn, or fluorescence signal. In the X-exe, PCR amplification cycle. Colors indicate different 10-fold dilution of *R.equi* DNA (ng/µL).

**Figure 3 animals-12-01172-f003:**
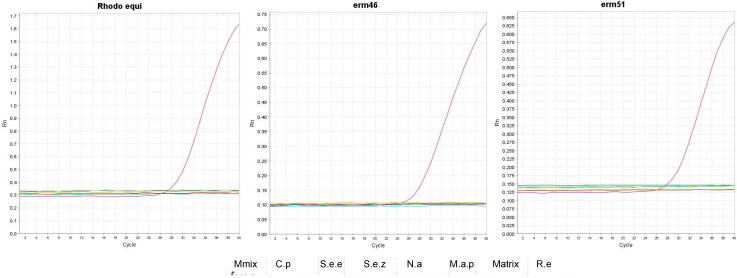
Testing the analytic specificity of Rhodo_Dlab, Erm46_Dlab, and Erm51_Dlab sets in the multiplex. qPCR curves for the three oligo sets in multiplex assays. In the Y-exe ΔRn, or fluorescence signal. In the X-exe, PCR amplification cycle. Colors indicate different bacteria species and PCR controls: pink- *R.equi* mix DNA (R.e); purple- nasal swab DNA (matrix); dark blue, *M. avium* subsp. *paratuberculosis* (M.a.p); light blue- *N. asteorides* (N.a); dark green-*S. equi* subsp. *zooepidermicus* (S.e.z); light green-*S. equi* subsp. *equi* (S.e.e); yellow- *C. pseudotuberculosis* (C.p); red negative control (no DNA, Mmix).

**Table 1 animals-12-01172-t001:** Oligos and probes.

Name	SEQUENCE 5’-3’	Product Size	Purpose
Rhodo_Dlab_F	TGTCAACAACATCGACCAGGC	200 bp	Amplifies *choE* gene, chromosomal marker in *R. equi*
Rhodo_Dlab_R	GCGTTGTTGCCGTAGATGAC		
Rhodo_Dlab_P	/56-FAM/CCGCCCAAC/ZEN/GTTCGGGTTTCACAACCGCTT/3IABkFQ/ *		
Erm46_Dlab_F	GTGGCGCAACGATGATGACT	192 bp	Amplifies macrolide resistance gene *erm46*
Erm46_Dlab_R	TGAAGACGGTGTGGACGAAG		
Erm46_Dlab_P	/5HEX/CCGCATCGG/ZEN/CGTTCACACCACGGC/3IABkFQ/ *		
Erm51_Dlab_F	CTGCCGTTTCACCTGACCAC	198 bp	Amplifies macrolide resistance gene *erm51*
Erm51_Dlab_R	GGGACGGAAATGTGTGGATG		
Erm51_Dlab_P	/5Cy5/GCCGGCGTC/TAO/GGTGGTGCCACGATGATGA/3IAbRQSp/ *		

* /56-FAM/-excitation 495 emission 520;/5HEX/- excitation 538 emission 555; 5Cy5/-excitation 648 emission 668.

**Table 2 animals-12-01172-t002:** In-silico validation of the oligos and probes.

	Match with Rhodo_Dlab Set				Match with Erm46_Dlab Set				Match with Erm51_Dlab Set			
	FW	RV	Probe	PCR Products	FW	RV	Probe	PCR Products	FW	RV	Probe	PCR Products
Macrolide Susceptible [*n* = 88]	88	88	88	87	1	88	1	0	1	0	88	0
Macrolide Resistant *erm*(46)-positive [*n* = 85]	85	85	85	83	85	85	85	85	2	1	85	0
Macrolide Resistant *erm*(51)-positive [*n* = 29]	29	29	29	29	0	29	14	0	29	29	29	29
Non- *R. equi* species [*n* = 24]	2	6	2	0	4	6	3	0	1	0	10	0

In hard brackets is the number of strains in each resistant genotype category. FW refers to the number of genomes that had a match with the forward oligo of the set. RV refers to the number of genomes that had a match with the reverse oligo of the set. Probe refers to the number of genomes that had a match with the probe of the set. PCR product indicates the number of genomes, in which the PCR set would produce a detectable PCR product (forward and reverse primers are in <2 kb proximity and corresponding probe is inside that <2 kb DNA fragment). This was calculated based on QuantiNova DNA polymerase processivity (Taq DNA Polymerase 2–4 kb/min) and PCR elongation time (30 s).

**Table 3 animals-12-01172-t003:** Efficiency, coefficient of determination (R^2^) and limit of determination (LOD) for each primer set in singleplex and multiplex assays.

	Efficiency (%)			R^2^			LOD		
	Singleplex	Multiplex	Multiplex Mocking *	Singleplex	Multiplex	Multiplex Mocking *	Singleplex	Multiplex	Multiplex Mocking *
Rhodo_Dlab	121.1	112.8	115.5	0.9976	0.9987	0.9904	10.7	11	10.7
Erm46_Dlab	104.7	105.5	102.8	0.9999	0.9994	0.9976	1.18	11.8	10.7
Erm51_Dlab	120.5	105.4	90.2	0.9993	0.9971	0.9917	1.07	10.7	106.9

LOD expressed as the minimal number of complete genome copies detected per PCR reaction. * Multiplex qPCR performed in the mocking equine respiratory samples spiked with *R. equi* DNA.

## Data Availability

The datasets analyzed for this study can be found in GenBank (https://www.ncbi.nlm.nih.gov/genbank/, accessed on 22 April 2022).

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
