# Peer review of "Novel Quantitative PCR for Rhodococcus equi and Macrolide Resistance Detection in Equine Respiratory Samples"

_animals, 2022, doi:10.3390/ani12091172_

Round 1

Reviewer 1 Report

I have reviewed the corrected manuscript and the authors comments. 

Here my comments:

Rephrasing the sentences on lines 75-80 and 159-166 did not answer the question about the specificity of primers and probes used in the study. This reviewer did BLAST search of all primers and probes listed in Table 1 as well as corresponding amplicons revealed 100% homology with other Rhodococcus strains, as well as other bacteria such as Spirosoma montaniterraeArthrobacter sp., Brevibacterium sp.

Melting temperature of assays used as singleplex and multiplex qPCR remain the main issue. Changing the melting temperature to those used in a preliminary study that used a conventional PCR is not scientifically correct. Stating that higher melting temperature won’t have significant detrimental effect on the PCR performance is not correct.

To assure accurate and trusted diagnostic results it is necessary to assess the nucleic acid extraction efficiency. The author’s statement: “Our assay is meant to be used for the recognition of R. equi and resistant genes in clinical settings and therefore the nucleic acid extraction methods are going to be very similar to the ones used in our study,” is not correct and does not justify the reason of not assessing the nucleic acid extraction efficiency.

There are few authors statements such as: a) reduction of primers concentrations, b) use of controls, have not properly addressed.

Based on the above comments this reviewer thinks that the manuscript is not improved and should not be considered for publication.

Reviewer 2 Report

The revised manuscript addresses all comments sufficiently.  The enhanced description of the in silico analysis and discussion on the genes being detected are very helpful.  

Author Response

We are happy to have satisfied all the reviewer's comments and we thank the reviewer for the feedback on our manuscript.

Reviewer 3 Report

The manuscript presents a new technique to detect Rhodococcus equi and its resistance to antimicrobials in just one qPCR (multiplex qPCR) decreasing the time of diagnosis and antimicrobial resistance.

Despite the effort made by the authors to determine the specificity of their new qPCR technique with clinical isolated samples, they didn´t use real samples to  determine the impact of their research which is the biggest handicap of the manuscript. The mock samples are not enough to understand  how this method can work with real sick animals.  For that reason, the results presented here should be considered as preliminar.

However, the results presented are good and probably they can be used with real clinical samples but for now is not possible confirm that.

Other points that should be addressed are:

The authors are using 34 non-R.equi strains instead of 24.

The annealing of the Erm46_Dlab primers are low in all Tm and they present several non-specific bands. Moreover,  seems that the best Tm is higher than 60ºC.

Despite the fact that R. equi is an emerging pathogen with an increase number of  antimicrobial resistance in its genome, the use of a farm in Kentucky as an example of the increasing number of antimicrobial resistances is not accurate. Moreover,  the presence  of R. equi strains is endemic of different farms. (Discussion section, Lines 247-248).

Round 2

Reviewer 1 Report

The authors did improve the manuscript that warrant publication in Animals. Just to mention, relying solely on the number and position of primer–template mismatches is insufficient and often misleading for in silico predictions of primer specificity. Primer specificity always increased in response to three factors: total NCM, total CM, and annealing temperature. In silico analysis is not sufficient to predict in vitro specificity. As such care must be taken when choosing the primers for a detection assay.

In addition, annealing temperature had substantial influence on primer specificity. It should be noted that direct increase in annealing temperature beyond the optimal level can reduce PCR efficiency, implying a reduction in the detection limit of the primer assay for DNA studies.

Author Response

We thank the reviewer for all the time invested in reviewing our manuscript. We believe his contribution helped to significantly improve our work.

Reviewer 3 Report

The authors amended all my concerns

Author Response

We are very thankful for the reviewer's contribution and happy that we address all the reviewer's concerns.

This manuscript is a resubmission of an earlier submission. The following is a list of the peer review reports and author responses from that submission.

Round 1

Reviewer 1 Report

The authors described a multiplex qPCR for detection of R. equi and related macrolide resistance genes. The main aim of this study was to develop and standardize a novel multiplex qPCR to detect R. equi and its most clinically relevant antimicrobial resistance genes directly from equine respiratory samples. R. equi is the most common cause of severe infectious pneumonia in foals. Current diagnostic approach includes bacterial culture combined with cytological examination and qPCR. The wide use of antimicrobials has led to development of macrolide resistance of R. equi so early detection of resistant strains is essential for adequate treatment. The authors developed and validated multiplex qPCR targeting a fragment of choE gene that encode cholesterol oxidase, and two resistant genes erm(46) and erm(51). The approach in developing multiplex qPCR described in the materials and methods of the study is adequate but have some flaws so there are several important points that require clarification and additional adjustment.

Points that need clarification:

  • All three assays for multiplex qPCR targeting different genes are not specific only for R. equi. There is a 100% homology with other Rhodococcus strains, as well as other bacteria such as Spirosoma montaniterrae, Arthrobacter sp., Brevibacterium sp. Such assays will definitively amplify other Rhodococcus strains as well as other bacteria. The authors did state (lines 132-135) that their: “primers set matches non-targeted R. equi sequences, amplification was never expected because the combination of the two oligos and corresponding probe was never found in enough proximity to generate a PCR product”. This statement is not current and needs to be rephrased or deleted.
  • The melting temperatures of each labeled probes ranges from 71-73.90C, which is above the recommended melting temperatures of 65-670 Higher melting temperature will impact the amplification efficiency and sensitivity.
  • In multiplex qPCR there is a variation in measuring the reference spectra and the sample spectra which introduces some error into the determination of each dye's spectra. The greater the spectra overlap between dyes, the greater the error. In addition, reactions to amplify multiple different segments in the same tube share common reagents. For accurate amplification, it is important that the multiple reactions do not compete. Competition can be avoided by limiting the concentration of primers used in the amplification reactions. It does not seem that the authors used such approach. The validation and optimization of multiplex qPCR is a very thorough process, and it is not just mixing different assays together.
  • What was the quencher used in the study?
  • The calculated qPCR efficiencies for two assays in singleplex and all three assays in multiplex were not acceptable. The acceptable efficiency is between 90-110%. Such high efficiencies suggest an inadequate assay design, inadequate validation and optimization, and inadequate baseline and threshold settings during analysis.
  • What was a positive and negative controls used?
  • A reference gene for biological samples such a swab should be a host conserved gene. This is necessary to assess nucleic acid extraction efficiency. The authors did not use it.
  • Negative results using mock samples does not prove specificity. The authors have a problem with a sensitivity, as efficiency of each assay is in an acceptable range.

Specific comments:

  • Line 23, delete “quantitative”.
  • Lines 59-60, the link is not valid.
  • Table 1 is not necessary. In the present form is confusing, so suggesting making a summary of it.

Reviewer 2 Report

Novel Quantitative PCR for Rhodococcus equi Diagnosis and ‎‎Macrolide Resistance Detection in Equine Respiratory Samples ‎by Narvaez et al is reviewed.  The manuscript describes development of a multiplex real time taqman based PCR to detect R. equi along with macrolide resistance genes. 

General Comments:

The approach is generally sound and would be useful for equine clinicians and diagnostic labs testing these types of samples.  The manuscript could improve upon describing the challenges with macrolide resistance a bit more.  For example what are the clinical outcomes with these strains?  Are their clinical breakpoints established, etc.   The major weakness of the manuscript is lack of clinical samples to validate the assay on, the entire approach is in silico.  Additionally, it is unclear how the set of reference sequences to test the targets in silico was assembled/developed and is confusing.  There is no section in the methods describing how this was done.  Also  Macrolide resistance genes are present in a wide variety of organisms, not just R. equi, and the authors should perform a more robust analysis to determine the relationship between detection of macrolide resistance and likelihood of isolation of a resistant strain to determine if potential reservoirs or other environmental organisms may cause false positive macrolide detection results. 

Specific Suggestions:

Title: Since there are no clinical samples included, I am not sure how you can use it for diagnosis.  Diagnosis needs clinical and additional information rather than just gene detection.  Also the sample set was DNA extracted from equine respiratory samples, not the samples themselves. 

Summary Abstract: Please define "oligo".  I assume this is oligonucelotides. 

Line 16: Clinical diagnosis is not done by a PCR test, these can inform this but they require additional information.  Detection is more descriptive.

Line 26:  should specify this is technical sensitivity not clinical

Line 28 what are units for 90 and 147

Line 29: any cross-reaction with normal flora...this should be qualified normal flora it was tested against.

Line 39: Cite economic costs? Could it be elaborated how this might be treated if it is resistant?  What are clinical outcomes of infections with a resistant strain.

Line 51: Does this assay actually identify macrolide resistant R. equi?  Or just the genes and the assumption is made these are from the same organism.  See comments above about this.  

Line 65: How was resistance determine in these strains?  Are their clinical breakpoints for these drugs?  Ecoffs? 

Line 78- How were these off target isolates identified/confirmed? 

Line 96 how were these Ct cutoffs estimated?  Automaticially?  Was there a specific threshold?

Line 132-137: There should be methods on how this assessment was performed.  How were the collections of sequences used?  Why was the non-R. equi set limited to mostly Rhocococcus sp. and a few others?  Could not a more comprehensive comparison be performed using public sequence databases? 

Table 1: What quenchers were used for these probes, I am assuming they are taqman with quencher.  

Table 2: is very confusing.  I am not sure a table is the best way to summarize this information.  See comments above about putting this analysis in methods.

Table 3: Can be supplementary.

Figure 2 Is it possible to show the standard curves rather than amplification?  or both? (log10 copies on y-axis instead of cycles and x-axis Ct instead of fluorescence?)

Line 184-185: Wouldn't a more closely representative mock infection be to add resistant/susceptible bacteria to the swab media and then extract?  This way all you're really looking at is DNA and not presence of inhibitors, other factors, etc.  This would be a closer mimic to clinical samples than just mixing DNA.  

Line 217: Wouldn't an antibiogram potentially be available since it is a population based summary?  Do you mean MIC?

Line 220: I am not sure this reflects "all" cases, just in the tested population

Line 223: This report appears to be from a single farm, not farms, it may not be representative. 

Line 237: Again, if this test is being run on clinical samples, how would you know the R. equi is carrying erm?

Reviewer 3 Report

The paper presented by Álvarez-Narváez showed important results to analyse the presence of Rhodococcus equi in equine samples, which will reduce the diagnosis time and also the efficiency of the treatments. However, with the results presented here, the authors could not demonstrate if their treatment can be useful with clinical samples. So, the main weaknes of the present study is that the the authors did not analyse the presence of R. equi samples in vivo therefore, the data should be considered as preliminar.

Moreover, I would like to highlight some minor points that could increase the overall quality of the manuscript. 

In the introduction, authors should increase the information on the difficulty  to identify R. equi especially in clinical samples where R. equi can be confused with other bacteria, especially with Dietzia genus.

In the same line, authors should analyse the specificity of the primers against Dieztia spp. which is a common cause of miss-diagnosis of R. equi.

The authors should describe MLSB in the simple summary where it appears the first time.

The primer forward to identify choE gene expression by qPCR is the same (except 1 bp) that the use by Ladron et al., 2003, the authors should highlight that when they presented this results 

Author Response

REVIEWER 3

The paper presented by Álvarez-Narváez showed important results to analyse the presence of Rhodococcus equi in equine samples, which will reduce the diagnosis time and also the efficiency of the treatments. However, with the results presented here, the authors could not demonstrate if their treatment can be useful with clinical samples. So, the main weaknes of the present study is that the the authors did not analyse the presence of R. equi samples in vivo therefore, the data should be considered as preliminar.

We thank the reviewer for this feedback. We are aware of the limitations of our work. Future work will address our current luck of clinical samples.

Moreover, I would like to highlight some minor points that could increase the overall quality of the manuscript. 

  • In the introduction, authors should increase the information on the difficulty to identify equi especially in clinical samples where R. equi can be confused with other bacteria, especially with Dietzia genus.

We thank the reviewer for this comment. We have expanded the introduction adding a small paragraph detailing the challenges involved in the diagnosis of R. equi (“Nevertheless, the identification of R. equi directly from respiratory samples can be challenging due to the changing morphology of this bacterium (from rods to cocci) and its high phenotypical similarity to other Actinobacteria such as Corynebacterium8 and Dietzia 9,10.” Lines 50-53)

  • In the same line, authors should analyze the specificity of the primers againstDieztia spp. which is a common cause of miss-diagnosis of  equi.

We thank the reviewer for this comment. We have performed the in-silico analysis in Dietzia aerolata reference genome ASM1414486v. Results are summarized in table 3.

  • The authors should describe MLSBin the simple summary where it appears the first time.

We thank the reviewer for this observation, we have corrected the manuscript accordingly (line 14)

  • The primer forward to identify choE gene expression by qPCR is the same (except 1 bp) that the use by Ladronet al., 2003, the authors should highlight that when they presented this results 

We highly thank the reviewer for this observation, we did not realize about the similarity, we used an automated oligo design software to design the oligos. We have cited authorship accordingly on the table.
